# A Low-Noise CMOS Transimpedance-Limiting Amplifier for Dynamic Range Extension

**DOI:** 10.3390/mi16020153

**Published:** 2025-01-28

**Authors:** Somi Park, Sunkyung Lee, Bobin Seo, Dukyoo Jung, Seonhan Choi, Sung-Min Park

**Affiliations:** 1Division of Electronic and Semiconductor Engineering, Ewha Womans University, Seoul 03760, Republic of Korea; 1118som@ewhain.net (S.P.); snklee20@ewha.ac.kr (S.L.); qhqls28@ewha.ac.kr (B.S.); seonhan.choi@ewha.ac.kr (S.C.); 2College of Nursing, Ewha Womans University, Seoul 03760, Republic of Korea; dyjung@ewha.ac.kr

**Keywords:** active feedback, APD, CMOS, LiDAR, limiting, transimpedance

## Abstract

This paper presents a low-noise CMOS transimpedance-limiting amplifier (CTLA) for application in LiDAR sensor systems. The proposed CTLA employs a dual-feedback architecture that combines the passive and active feedback mechanisms simultaneously, thereby enabling automatic limiting operations for input photocurrents exceeding 100 µA_pp_ (up to 1.06 mA_pp_) without introducing signal distortions. This design methodology can eliminate the need for a power-hungry multi-stage limiting amplifier, hence significantly improving the power efficiency of LiDAR sensors. The practical implementation for this purpose is to insert a simple NMOS switch between the on-chip avalanche photodiode (APD) and the active feedback amplifier, which then can provide automatic on/off switching in response to variations of the input currents. In particular, the feedback resistor in the active feedback path should be carefully optimized to guarantee the circuit’s robustness and stability. To validate its practicality, the proposed CTLA chips were fabricated in a 180 nm CMOS process, demonstrating a transimpedance gain of 88.8 dBΩ, a −3 dB bandwidth of 629 MHz, a noise current spectral density of 2.31 pA/√Hz, an input dynamic range of 56.6 dB, and a power dissipation of 23.6 mW from a single 1.8 V supply. The chip core was realized within a compact area of 180 × 50 µm^2^. The proposed CTLA shows a potential solution that is well-suited for power-efficient LiDAR sensor systems in real-world scenarios.

## 1. Introduction

Light detection and ranging (LiDAR) technology has become an essential tool in short-range applications such as indoor navigation, fall detection, and monitoring systems [1,2,3,4,5,6]. These applications demand low-power and cost-effective solutions, making LiDAR sensors indispensable in modern short-range detection systems. As the adoption of LiDAR sensors continues to grow, the need for compact and efficient circuit designs that ensure reliable performance while satisfying stringent power and size constraints has become increasingly critical.

In a typical LiDAR system, laser pulses are reflected from objects and captured by an avalanche photodiode (APD) which translates the received optical signals into the equivalent electrical currents. These signals are amplified by the front-end transimpedance amplifier (TIA, as commonly defined in optical sensors and systems), boosted further by the following post-amplifier (or frequently, a multi-stage limiting amplifier to extend the dynamic range), and then processed by a time-to-digital converter to determine the distance to targets, as shown in Figure 1a. Among these several blocks, the TIA circuit positioned at the front of the optical receiver chain plays a crucial role in amplifying weak input current signals, while minimizing the noise and distortion phenomena. Therefore, achieving high transimpedance gain, wide bandwidth, low noise, and low power consumption simultaneously is vital to the overall system performance of TIAs [7,8,9,10].

Despite the various advantages of a conventional shunt-feedback TIA, it often faces considerable limitations, including gain–bandwidth tradeoffs and restricted dynamic range. In particular, the latter necessitates an additional component such as automatic-gain-control (AGC) circuitry. Consequently, this leads to circuit complexity and increased power consumption. To alleviate these design challenges, the proposed TIA circuit in this paper employs a dual-feedback configuration instead of the traditional shunt-feedback approach. Nonetheless, this modification still suffers from pulse-signal distortions, thus mandating further enhancements. Hence, we incorporate an active feedback loop with a carefully tuned resistor (R_F2_), which can then successfully mitigate the signal distortions and enhance the circuit’s stability as well. A diode-connected NMOS switch is inserted at the input of the active feedback loop for this purpose, i.e., to activate the feedback loop only when the input photocurrent exceeds 100 µA_pp_.

Another notable aspect of this design approach lies in its capacity to drive output pulses rapidly into saturation for input currents exceeding 100 µA_pp_, which results in the great enhancement of the circuit dynamic range. This saturation (or so-called limiting) mechanism certainly eliminates the following multi-stage limiting amplifier, thereby reducing the overall power consumption and the system complexity concurrently as depicted in Figure 1b.

In Section 2, the circuit operations are discussed along with the on-chip photodiode design. In Section 3, the chip layout and simulation results are presented. Section 4 outlines the measurement data of the fabricated chip, and Section 5 concludes this work.

## 2. Circuit Description

Figure 2a presents the block diagram of a frequently utilized shunt-feedback TIA (SF-TIA) which consists of a three-stage inverter-based amplifier and an automatic-gain-control (AGC) circuit equipped with an adjustable feedback resistor. It is then followed by a multi-stage limiting amplifier to extend the input dynamic range. Shunt-feedback TIAs have been widely adopted due to their simplicity and reasonable performance [11,12,13,14,15]. However, their usage can be limited by several issues such as the inherent tradeoff between gain and bandwidth, as well as their restricted dynamic range. Therefore, additional components such as AGC circuits are required. In particular, the voltage gain of the inverter amplifiers plays a key role in determining the transimpedance gain. Yet, the voltage gain should be carefully optimized to maintain the circuit stability. Therefore, the three-stage architecture is typically preferred to achieve a relatively high voltage gain while ensuring stable operations. In contrast, Figure 2b shows the simplified block diagram of the proposed CMOS transimpedance-limiting amplifier (CTLA) that features the same three-stage inverter amplifier with a feedback resistor (R_F1_), an active feedback TIA with R_F2_, and an NMOS switch. It should be noted that the NMOS switch is strategically positioned between the input node and the active feedback TIA so that the feedback loop can be activated only when the amplitude of the input current exceeds 100 µA_pp_. In other words, the active feedback TIA contributes to the rapid saturation of the output signals, hence effectively eliminating the need for a power-hungry limiting amplifier in its subsequent stage.

### 2.1. Main TIA with Dual-Feedback and Multi-Stage Inverters

The proposed CTLA employs a dual-feedback topology, in which the main feedback resistor (R_F1_) connects the output (v_o1_) to the input node while the local feedback resistor (R_F3_) is placed at the third inverter stage. Figure 3a depicts the schematic diagram of the main dual-feedback TIA (DF-TIA) that consists of a three-stage inverter chain and a feedback resistor. Figure 3b compares the simulated frequency response of the DF-TIA with that of the conventional SF-TIA. When evaluated under the same bandwidth characteristic, the DF-TIA demonstrates an approximately 4 dB higher transimpedance gain. This superior gain of the DF-TIA can be exploited to achieve a higher amplification in the CTLA, while maintaining the pulse integrity and stability despite the active feedback path.

### 2.2. Active Feedback TIA and Output Buffer

Although the DF-TIA architecture achieves a higher transimpedance gain when compared with a conventional SF-TIA design, it might still suffer from pulse distortions even for input currents below 100 µA_pp_, consequently leading to a limited dynamic range. To overcome this issue, an active feedback inverter TIA with a feedback resistor (R_F2_) is employed in this work, as shown in Figure 4. This novel topology enables the maintenance of undistorted output signals even under the conditions of large input photocurrents. Its operation principles are described in detail below.

First, an NMOS switch is inserted in the feedback path as a voltage-controlled component. It toggles automatically with an input current threshold of 100 µA_pp_, and therefore the DF-TIA performances can be preserved for lower input currents. However, for higher input currents than the threshold, the active feedback TIA is activated and hence compensates for distorted pulses and thus helps to expand the input dynamic range. With these dual-mode operations, the power-hungry limiting amplifier can be discarded. In short, this simple design methodology can offer a low-power, cost-efficient alternative for use in short-range LiDAR sensors and optical interconnect systems.

Second, the value of R_F2_ in the active feedback TIA is carefully optimized because an increase in R_F2_ can amplify the feedback inverter’s effect and thus give rise to instability or oscillations. Simultaneously, the voltage gain (A_2_) of the active feedback path must be judiciously tuned to ensure circuit stability.

The small-signal analysis of the proposed CTLA provides the following expressions for the input resistance and the transimpedance gain, respectively. More specifically, Equation (1) describes the condition when the active feedback TIA is inactive, while Equation (2) corresponds to the scenario where the active feedback TIA is activated.(1)$$R_{in}=\frac{v_{in}}{i_{in}}=\frac{R_{F1}}{1+A_{1}},\;Z_{T}=-R_{F1}\left(\frac{A_{1}}{1+A_{1}}\right)$$(2)$$R_{in}=\frac{v_{in}}{i_{in}}=\frac{R_{F}}{1+\alpha A_{1}}=\frac{\left(R_{F1}{||R}_{F2}\right)}{1+\alpha A_{1}},\;Z_{T}=-R_{F}\left(\frac{A_{1}}{1+\alpha A_{1}}\right)=-(R_{F1}|\left|R_{F2}\right)\left(\frac{A_{1}}{1+\alpha A_{1}}\right)$$
where A1 denotes the voltage gain of the DF-TIA and $\alpha ≅1-(g_{m7}+g_{m8})R_{F}$.

It is clearly seen that the input resistance (R_in_) can be significantly reduced by the parallel combination of R_F1_ and R_F2_, as far as the value of R_F2_ becomes small enough. This approach effectively preserves the input photocurrents, while also enabling the extension of the −3 dB bandwidth of the proposed CTLA.

Figure 5a illustrates the MATLAB simulation results for the variations of the input resistance (Rin) and the transimpedance gain (ZT) with respect to the changes of RF1 and
α values. The top two plots correspond to the results of Equation (1), while the bottom plot represents the results of Equation (2). In this plot, the transimpedance gain of the CTLA is reduced to 35 dBΩ owing to the active feedback mechanism, which therefore allows the CTLA to tolerate large input currents exceeding 100 µApp and helps to recover output pules with no significant distortions.

Third, an inverter-based output buffer (I-OB) is utilized to align the output impedance to the standard value of 50 Ω and to efficiently decouple the load capacitance at the output stage of the CTLA. Figure 5b shows the schematic diagram of the I-OB, comprising a primary inverter stage followed by an additional inverter-configured TIA designed to obtain the impedance matching. Post-layout simulations indicate that the output resistance is approximately 67.5 Ω.

**Figure 5 micromachines-16-00153-f005:**
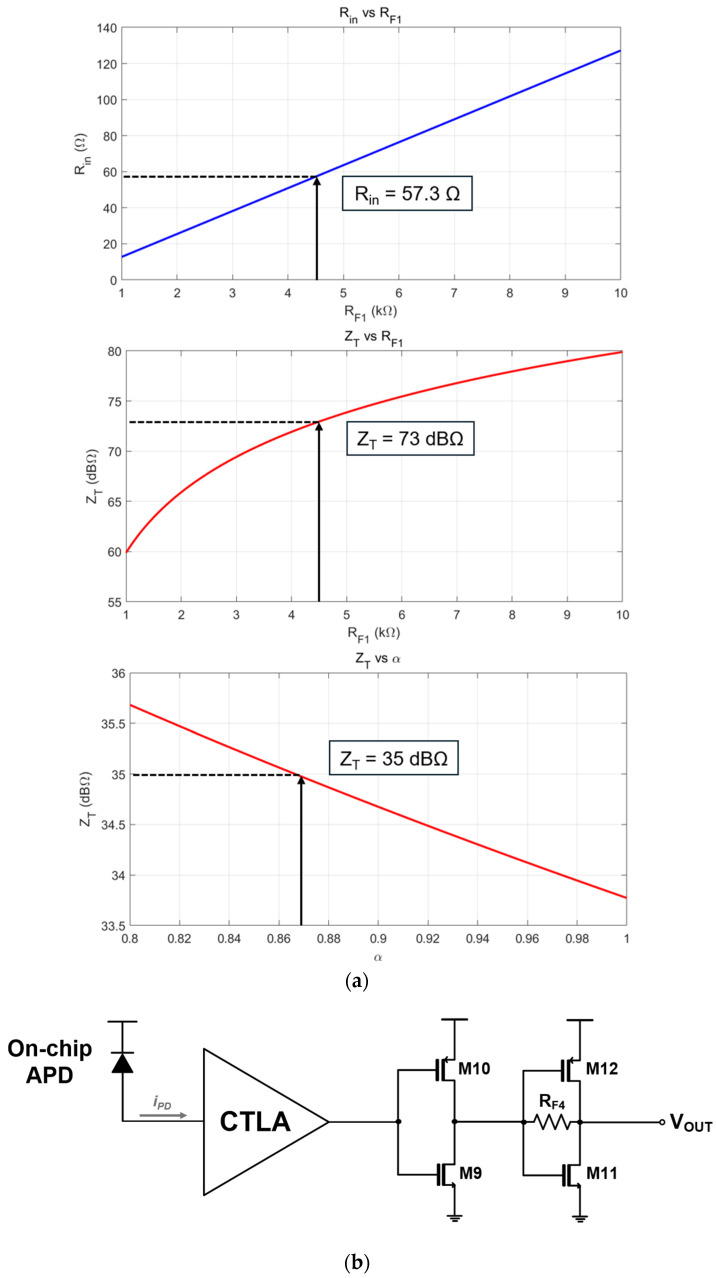
(**a**) Variation of the input resistance and the transimpedance gain with respect to the values of R_F1_ and α, and (**b**) schematic diagram of the I-OB.

### 2.3. On-Chip P^+^/NW/DNW APD

Figure 6a depicts the cross-sectional structure of the CMOS P^+^/N-well/Deep N-well (P^+^/NW/DNW) APD. It is well known that avalanche multiplication occurs at the P^+^/NW junction, and that the shallow trench insulators (STIs) prevent the notorious edge breakdown. The addition of the DNW layer enhances near-infrared sensitivity by restricting the hole diffusion into the p-substrate, and the built-in potential barrier blocks the undesirable photocurrents generated from the p-substrate.

Figure 6b presents the layout of the on-chip P^+^/NW/DNW APD, where a salicide blocking layer forms an optical window. The P^+^ contacts in the center should be exposed for low resistivity [16]. The optical window has a diagonal length of 40 µm, which leads to a depletion capacitance of 470 fF and a photodetection bandwidth of 1.7 GHz under a reverse bias voltage of 10.25 V [17]. It should be noted that an octagonal shape is preferred because it can reduce the edge breakdown effect.

**Figure 6 micromachines-16-00153-f006:**
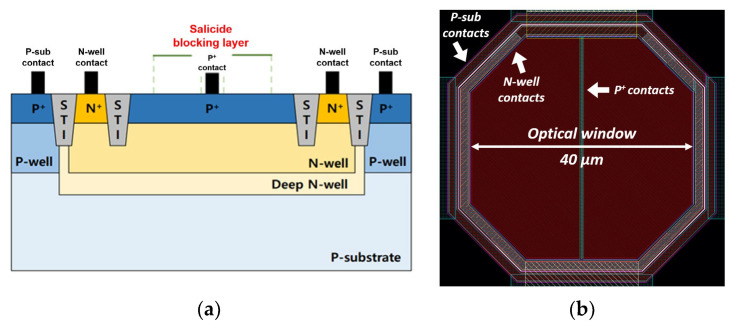
(**a**) Cross-sectional view of the P^+^/NW/DNW APD, and (**b**) layout of the on-chip APD.

## 3. Chip Layout and Post-Layout Simulation Results

Figure 7 shows the layout of the CTLA with an on-chip P+/NW/DNW APD, occupying an area of 180 × 50 μm^2^.

Post-layout simulations for the CTLA were carried out using the model parameters based on a standard 180 nm CMOS technology (TSMC, Taiwan). The DC simulation results indicate that the proposed CTLA consumes 23.6 mW from a single 1.8 V supply.

Figure 8a illustrates the frequency response of the CTLA, showing a transimpedance gain of 70.5 dBΩ and a bandwidth of 1.21 GHz for a photodiode capacitance of 0.47 pF extracted from the on-chip P^+^/NW/DNW APD. Additionally, the noise current spectral density is calculated to be 4.3 pA/√Hz for the simulated 1.21 GHz bandwidth.

Figure 8b reveals the post-layout simulation result of the phase margin (PM) that is approximately 60° for the utilized value of R_F2_. It confirms the stability of the proposed CTLA. However, as the feedback resistor R_F2_ increases, the PM decreases, thus highlighting that R_F2_ plays a critical role in determining the stability of the CTLA.

**Figure 8 micromachines-16-00153-f008:**
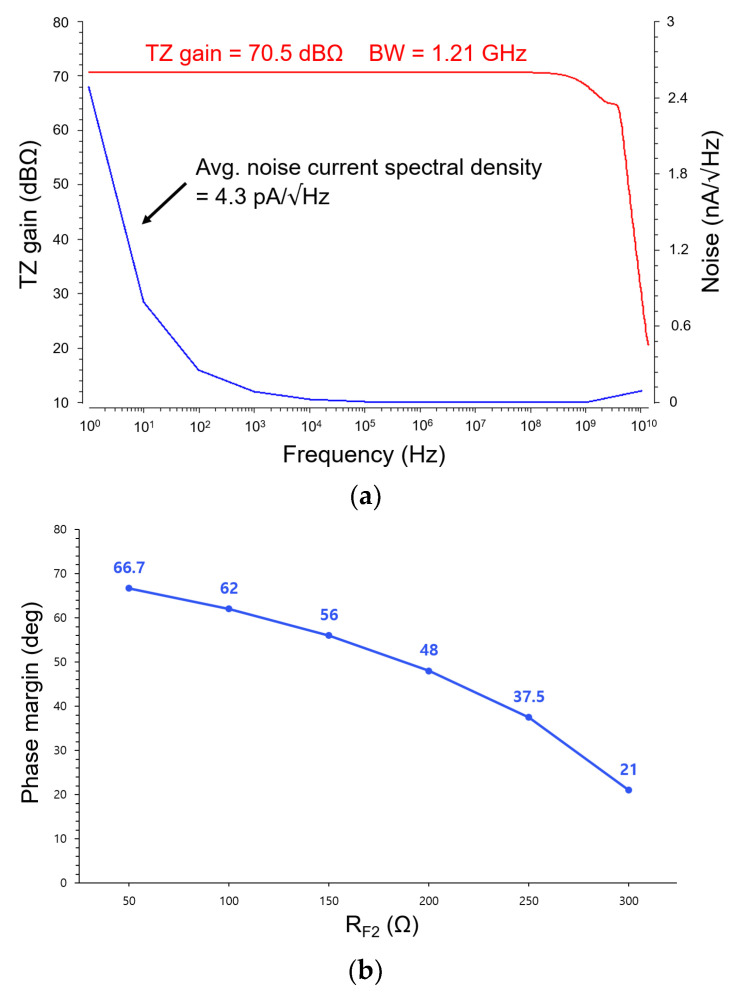
(**a**) Simulated frequency response (i.e., transimpedance gain, bandwidth, and noise current spectral density) and (**b**) phase margin characteristic of the proposed CTLA.

Figure 9 displays the simulated eye-diagrams of the CTLA at 300 Mb/s data rate for input currents ranging from 1 μA_pp_ to 1.5 mA_pp_, showing wide and clear eye-openings across this range. Although overshoots appear at the falling edges when the input currents exceed 500 μA_pp_, this might be attributed to the influence of the active feedback mechanism. Additionally, it is evident that the eye-heights reach saturation for input currents exceeding 100 μA_pp_, as anticipated.

Figure 10 depicts the simulated pulse response of the CTLA, in which the output pulse increases almost linearly for input currents smaller than 100 μA_pp_, whereas it saturates for larger photocurrents above 100 μA_pp_. This confirms the limiting behavior of the proposed CTLA for large input currents and its extended dynamic range.

Figure 11 compares the simulation results of the input current signals for the three TIA architectures, i.e., the proposed CTLA, the SF-TIA, and the DF-TIA, respectively. It becomes apparent that the conventional SF-TIA and the DF-TIA exhibit significant distortions at both rising and falling edges because of their limited dynamic range. In contrast, the CTLA delivers clean and undistorted pulse responses, thus validating its superior dynamic range characteristic even at the input node of the circuit.

Table 1 lists the performance deviations resulting from the process, voltage, and temperature (PVT) variations for three worst-case scenarios: (a) SS, 1.62 V, 125 °C, (b) TT, 1.8 V, 27 °C, and (c) FF, 1.98 V, −40 °C, respectively. Here, it is seen that the transimpedance (TZ) gain variation of the proposed CTLA is less than 7.5%, the bandwidth deviates by less than 4.13% from the nominal 1.21 GHz, and the noise current spectral density fluctuates by less than 18.6%. The transient response of the CTLA shows that the output voltage swing changes considerably for an input current of 1 μA_pp_. Additionally, the output varies within 24.6% for an input current of 100 μA_pp_. In addition, the DC voltage at the output node of the CTLA changes within 8.8% at the worst case. These simulation results certainly prove the stable operations of the proposed CTLA.

## 4. Chip Fabrication and Measured Results

Test chips of the proposed CTLA were fabricated in a standard 180 nm CMOS process. While smaller process nodes such as 28 nm or 65 nm offer advantages in terms of power efficiency and speed, their increased design complexity, process variability, and manufacturing costs often make larger nodes such as 180 nm and 130 nm more suitable for certain applications, especially those requiring high stability and lower design complexity [18]. In particular, the efficient photo-response of the on-chip APD mandates a smaller number of metal layers in CMOS processes, thus the preference for a 180 nm CMOS.

Figure 12 shows the chip micrograph and its test setup. The entire chip occupies the area of 1.0 × 2.0 mm^2^ including I/O pads, while the chip core covers the area of 180 × 50 μm^2^. The input signals are provided by the function generator (Tektronix AFG 31000, Beaverton, OR, USA) and the output signals are observed using an oscilloscope (Keysight DSOX1202A, Santa Rosa, CA, USA). It is noted that 10 samples of the CTLA chip were tested during the measurements.

Figure 13 demonstrates the measured frequency response of the proposed CTLA, demonstrating the transimpedance gain of 88.8 dBΩ and the −3 dB bandwidth of 629 MHz.

Figure 14 presents the measured output noise voltage of the CTLA with the Keysight EXR254A, acquiring the equivalent noise current spectral density of 2.31 pA/√Hz. Under the assumption of a BER of 10^−12^ and a responsivity of 4.16 A/W, this corresponds to an optical sensitivity of −40.1 dBm. Notably, the inherent noise voltage of the oscilloscope (Keysight DSOX1202A) was determined to be 0.266 mV_RMS_.

Figure 15 shows the measured eye-diagrams of the CTLA operating at 300 Mb/s for the 2^31^-1 PRBS input currents with test points at 165 µA_pp_, 330 µA_pp_, 665 µA_pp_, and 1.35 mA_pp_, respectively. Figure 16 demonstrates the measured eye-diagrams of the CTLA for the 2^31^-1 PRBS input current of 330 µA_pp_ at different data rates of 100 Mb/s, 300 Mb/s, 500 Mb/s, and 700 Mb/s, respectively. These measured results clearly demonstrate that the proposed CTLA maintains wide and clean eye-openings across a broad range of input currents and supports various data rates effectively.

Figure 17 displays the measured pulse response of the proposed CTLA for different input current levels of (a) 2 µA_pp_, (b) 100 µA_pp_, (c) 400 μA_pp_, and (d) 1 mA_pp_, respectively. Due to the limitation of the function generator, the minimum pulse width is set only to 10 ns. It is clearly seen that the CTLA results in clean output pulses, demonstrating the input dynamic range of 56.6 dB. Conclusively, the CTLA is capable of processing input photocurrents spanning from the minimal signal levels of 2 μA_pp_ to as high as 1.35 mA_pp_.

Table 2 compares the performance of the proposed CTLA with previously reported CMOS TIAs.

Ref. [19] proposed an analog front-end circuit tailored for pulsed TOF 4-D imaging LADAR applications. The design utilized a dual-topology TIA combining the shunt-feedback and current-mirror configurations to effectively handle both weak and strong signals, achieving a wide dynamic range. A double-threshold timing discriminator was also employed to ensure timing accuracy, thereby compensating walk errors and improving detection reliability. The circuit was fabricated in a 180 nm CMOS technology, showcasing its suitability for compact, low-power LADAR systems.

Ref. [20] presented a linear-mode LiDAR sensor system featuring a multi-channel CMOS TIA array paired with InGaAs PIN photodiodes. The design leveraged a voltage-mode feedforward TIA, offering twice the transimpedance gain and lower noise compared with a conventional inverter-based TIA, while maintaining a similar bandwidth. The system achieved robust detection of targets with a low reflectivity of 5% within 25 m range with the aid of the AGC mechanism. This implementation highlighted a cost-effective, low-power solution for LiDAR systems, effectively addressing the noise and sensitivity challenges with compact integration.

Ref. [21] introduced a precision TIA with minimal noise designed with a power-on-calibration (POC) technique to enhance the transimpedance gain precision and mitigate mismatches in multi-channel configurations. The proposed circuit was implemented in a 180 nm CMOS technology, featuring a transimpedance gain of 60 dBΩ, a bandwidth of 250 MHz, and a dynamic range of 66 dB. This design incorporated a noise-canceling method to achieve an equivalent input noise current density of 3.1 pA/√Hz. While the calibration mechanism ensured the reduction of gain deviation from ±40% to less than ±5%, the POC circuit’s reliance on a single off-chip resistor offered simplicity in implementation without excessive area overhead. This work demonstrated a significant improvement in sensitivity and gain stability, making it suitable for precise LiDAR receiver systems.

Ref. [22] introduced a CMOS fully differential optoelectronic receiver IC tailored for LiDAR solutions designed for near-field applications. The design integrated an on-chip CMOS P^+^/NW APD to mitigate signal distortion typically caused by the ESD protection diodes. The proposed architecture employed a dual-feedback folded-cascode TIA for enhanced input impedance reduction, an active single-to-differential converter for signal symmetry, and a two-stage differential amplifier with negative impedance compensation for gain and bandwidth optimization. The chip achieved a transimpedance gain of 87 dBΩ, a bandwidth of 577 MHz, and an input dynamic range of 50 dB while consuming 50.6 mW from a 1.8 V supply.

Although the proposed CTLA is originally built upon our prior work described in [23], we have expanded the analysis and simulations in this paper, along with the chip fabrication and its experimental results. With this silicon-proven data, we have demonstrated its potential for a low-noise low-power solution in the applications of short-range LiDAR sensors.

## 5. Conclusions

We have presented a novel CTLA realized in a 180 nm CMOS process for application in short-range LiDAR sensors, which incorporates an active feedback TIA with an NMOS switch not only to eliminate the need for a power-intensive multi-stage limiting amplifier, but also to lower the noise current spectral density further. With this circuit configuration, the CTLA can significantly improve power efficiency, and also extend the input dynamic range characteristic. Therefore, the proposed CTLA provides a low-noise, low-power, cost-effective solution well-suited for usage in short-range LiDAR sensor systems.

Meanwhile, the proposed CTLA requires fine-tuning processes during the circuit design, which are quite essential to guarantee optimal performance and provide flexibility across various application fields. Thus, it would require future work to reduce tuning sensitivity and explore automated tuning methods, thereby enhancing the usability and the scalability of the circuit. Despite these challenges, the proposed CTLA has potential to broaden its applications further and hence impact the fields of LiDAR technologies.

## Figures and Tables

**Figure 1 micromachines-16-00153-f001:**
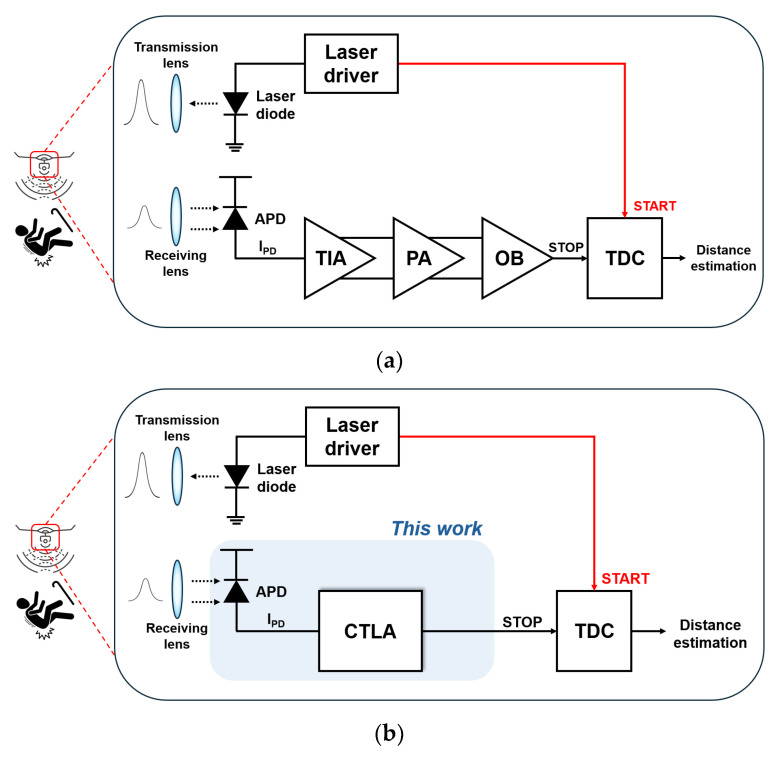
Block diagrams of (**a**) a typical LiDAR sensor, (**b**) the proposed LiDAR system.

**Figure 2 micromachines-16-00153-f002:**
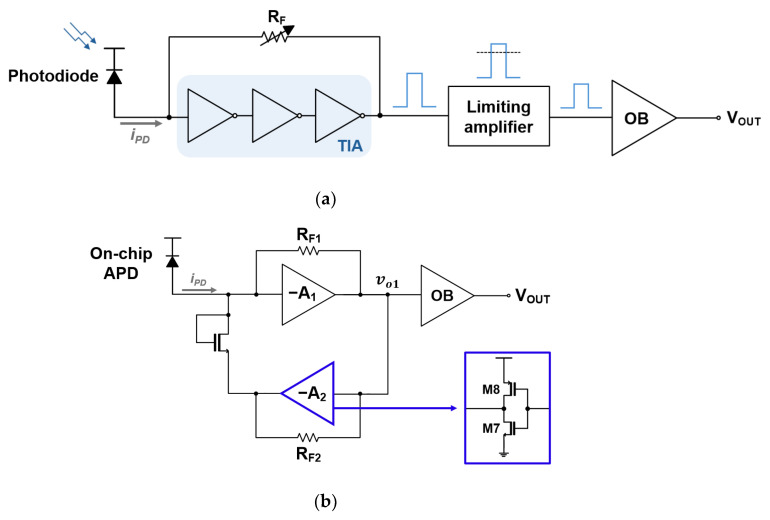
Block diagrams of (**a**) a conventional SF-TIA and (**b**) the proposed CTLA.

**Figure 3 micromachines-16-00153-f003:**
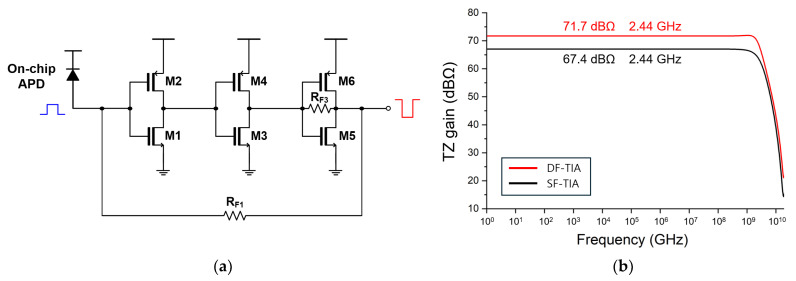
(**a**) Schematic diagram of the DF-TIA and (**b**) simulated frequency responses of the DF-TIA and a conventional SF-TIA for the same bandwidth.

**Figure 4 micromachines-16-00153-f004:**
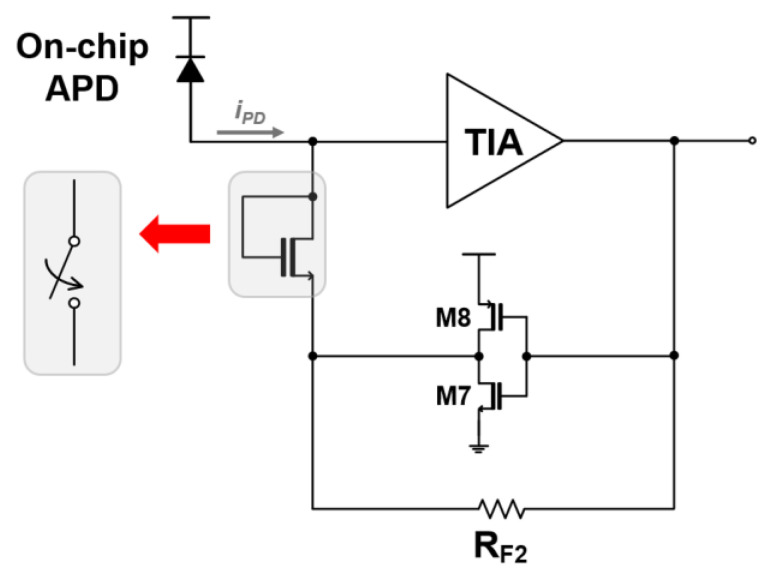
Schematic diagram of the inverter-based active feedback TIA.

**Figure 7 micromachines-16-00153-f007:**
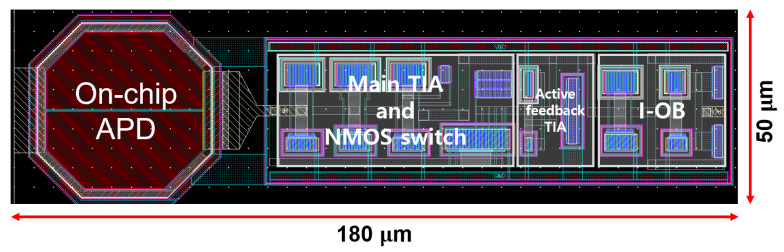
Layout of the proposed CTLA.

**Figure 9 micromachines-16-00153-f009:**
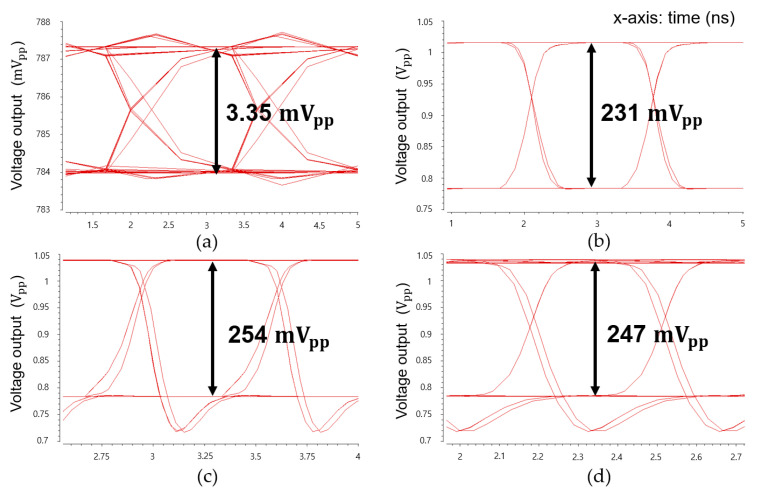
Simulated eye-diagrams of the CTLA at 300 Mb/s data rate with input currents of (**a**) 1 μA_pp_, (**b**) 100 μA_pp_, (**c**) 500 μA_pp_, and (**d**) 1.5 mA_pp_, respectively.

**Figure 10 micromachines-16-00153-f010:**
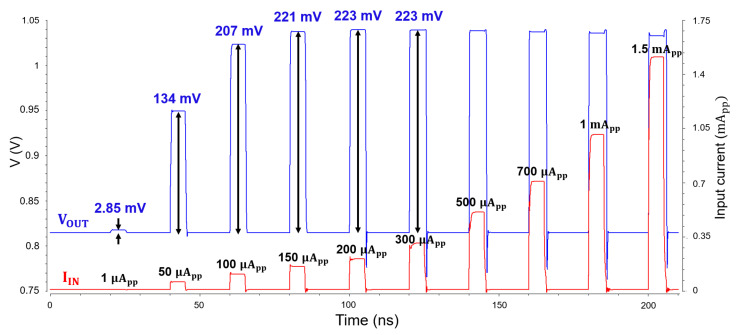
Simulated pulse response of the CTLA for various input currents.

**Figure 11 micromachines-16-00153-f011:**
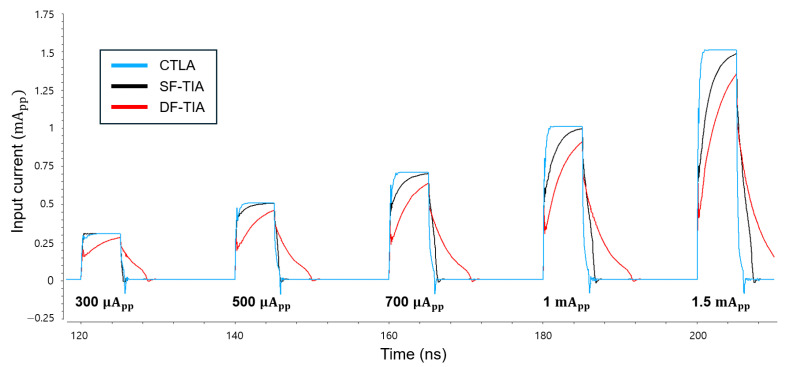
Simulated current pulses at the input nodes of the CTLA, SF-TIA, and DF-TIA (pulse width: 5 ns).

**Figure 12 micromachines-16-00153-f012:**
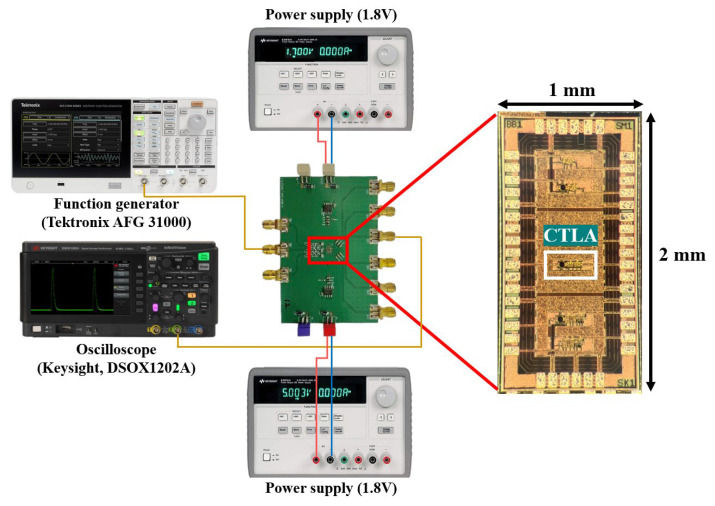
Chip photo of the proposed CTLA and its test setup.

**Figure 13 micromachines-16-00153-f013:**
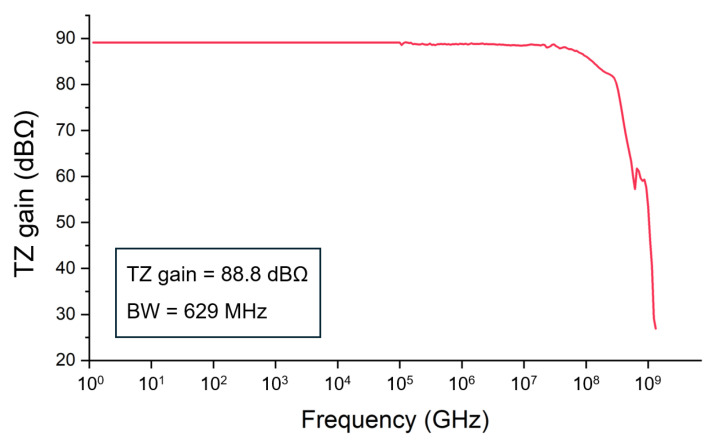
Measured frequency response of the CTLA.

**Figure 14 micromachines-16-00153-f014:**
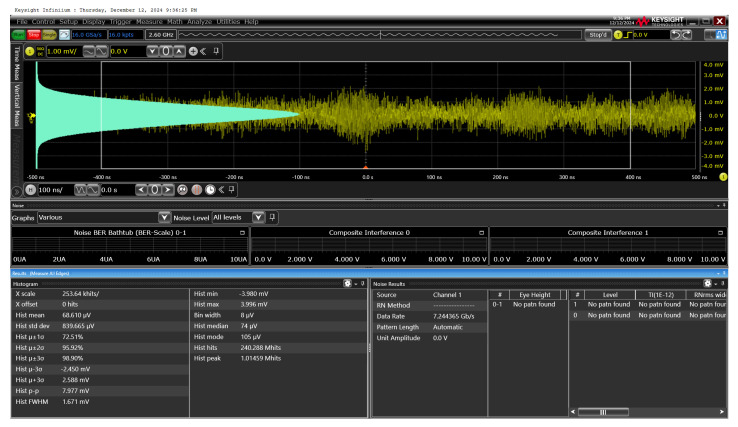
Measured output noise of the CTLA.

**Figure 15 micromachines-16-00153-f015:**
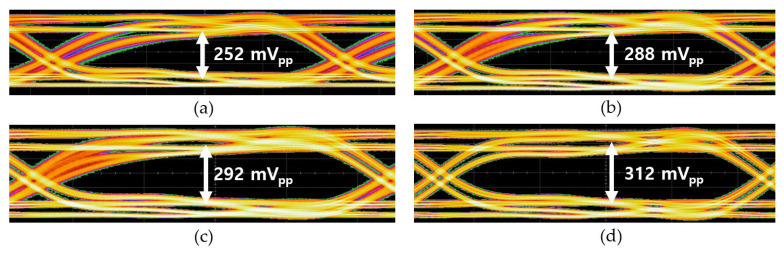
Measured eye-diagrams of the CTLA at 300 Mb/s data rate with the input currents of (**a**) 165 μA_pp_, (**b**) 330 μA_pp_, (**c**) 665 μA_pp_, and (**d**) 1.35 mA_pp_, respectively.

**Figure 16 micromachines-16-00153-f016:**
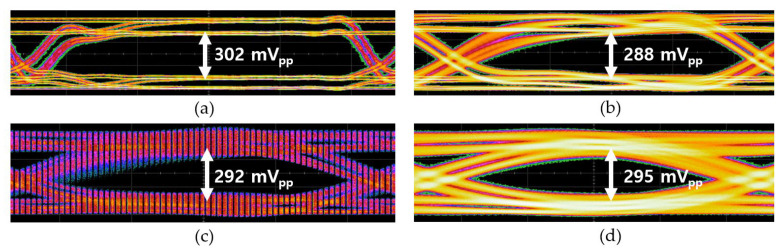
Measured eye-diagrams of the CTLA for the 2^31^-1 PRBS input current of 330 µA_pp_ at different data rates of (**a**) 100 Mb/s, (**b**) 300 Mb/s, (**c**) 500 Mb/s, and (**d**) 700 Mb/s, respectively.

**Figure 17 micromachines-16-00153-f017:**
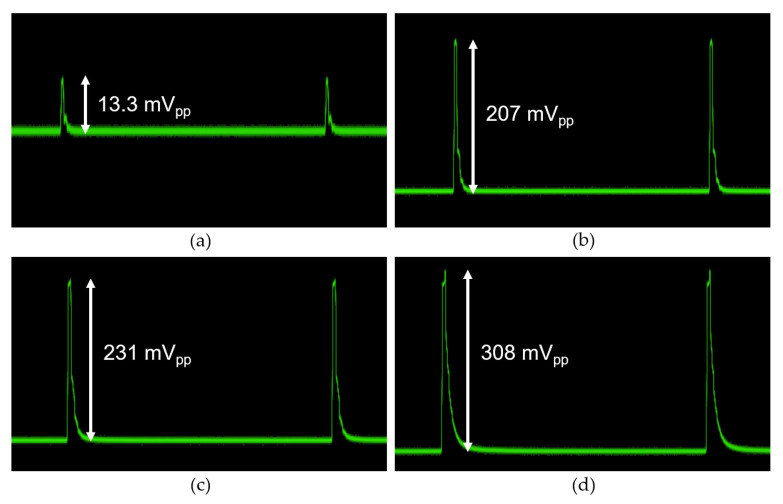
Measured pulse response of the CTLA for input currents of (**a**) 2 µA_pp_, (**b**) 100 µA_pp_, (**c**) 400 µA_pp_, and (**d**) 1 mA_pp_, respectively.

**Table 1 micromachines-16-00153-t001:** PVT simulation results of the proposed CTLA.

Parameters	SS, 1.62 V, 125 °C	TT, 1.8 V, 27 °C	FF, 1.98 V, −40 °C
TZ gain (dBΩ)	72.9 (+3.4%)	70.5	65.2 (−7.5%)
Bandwidth (GHz)	1.16 (−4.13%)	1.21	1.20 (−0.83%)
Noise current spectral density (pA/√Hz)	5.1 (+18.6%)	4.3	3.7 (−13.95%)
Output voltage amplitude (mV_pp_) @ 1 µA_pp_ input current	4.05 (+42.1%)	2.85	1.42 (−50.2%)
Output voltage amplitude (mV_pp_) @ 100 µA_pp_ input current	248 (+19.8%)	207	156 (−24.6%)
DC voltage V_o1_ (V) @ output node of CTLA	0.764 (−7.95%)	0.830	0.903 (+8.8%)

**Table 2 micromachines-16-00153-t002:** Performance comparison of the proposed CTLA with CMOS TIAs reported in earlier works.

Parameters	[19]	[20]	[21]	[22]	This Work
CMOS technology (nm)	180	180	180	180	180
Type	Off-chip (APD)	Off-chip (PIN)	Off-chip (APD)	On-chip (APD)	On-chip (APD)
C_pd_ (pF)	1.5	0.5	1.5	0.49	0.49
Responsivity (A/W)	45	0.9	-	2.72	4.16
Wavelength (nm)	905	1550	-	850	850
TZ gain (dBΩ)	106	76.3	60	87	88.8
Bandwidth (MHz)	153	720	250	577	629
Min. detectable current (µA_pp_)	0.54	1.14	0.5	4.18	2.0
Max. detectable current (mA_pp_)	5.5	1.0	1.0	1.33	1.35
Dynamic range (dB)	80	59.7	66	50	56.6
Power dissipation per channel (mW)	16.5 @3.3 V	29.8 @1.8 V	39.6 @3.3 V	50.6 @1.8 V	23.6 @1.8 V
Noise current spectral density (pA/√Hz)	0.89	6.3	3.1	15.4	2.31
Core area (mm^2^)	1.2 × 1.13 *	5.0 × 1.1 *	0.55 × 0.30	0.581 × 0.196	0.18 × 0.05

* I/O PADs included.

## Data Availability

Data are contained within the article.

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
