# Peer review of "A Low-Noise CMOS Transimpedance-Limiting Amplifier for Dynamic Range Extension"

_micromachines, 2025, doi:10.3390/mi16020153_

Round 1

Reviewer 1 Report

Comments and Suggestions for Authors

In the manuscript under review, the authors present a CMOS transimpedance-limiting amplifier based on a dual-feedback architecture for the LiDAR applications. The results of the analysis were obtained by simulation and experiment on a fabricated chip. The topic of the research is certainly relevant, and the results are interesting. The structure of the manuscript, the style of presentation, and the quality of the illustrations leave an overall positive impression. However, the following issues need to be further clarified for better publication needs.

1. Please provide more details about the simulation environment. Are all simulation results obtained in MATLAB?  Which frameworks have the authors utilized?

2. Can the authors show in Figure 10 the simulation results with the same test points as the experimental results in Figure 18? It might be necessary to discuss the correspondence between the simulation and experimental results.

3. Possible limitations and disadvantages of this study should be discussed in the conclusion.

4. In Table 1, there is an issue with the alignment of the first column.

Author Response

1. Please provide more details about the simulation environment. Are all simulation results obtained in MATLAB?  Which frameworks have the authors utilized?

(ans.) All simulations (including initial circuit design and simulations, layout and post-layout simulations) were conducted by using an EDA tool, i.e., Cadence Virtuoso.

2. Can the authors show in Figure 10 the simulation results with the same test points as the experimental results in Figure 18? It might be necessary to discuss the correspondence between the simulation and experimental results.

(ans.) Since the eye-diagrams are constructed by overlaying multiple pulse responses, the measured results may show slight variations from the simulations in the position of the lines. For the experiments, the open extent of the eyes was consistently measured using the y-marker function to ensure comparability. Despite the minor variations, the key metrics for comparison such as eye-opening align well between the simulation and the experimental results.

3. Possible limitations and disadvantages of this study should be discussed in the conclusion.

(ans.) We have discussed the limitations of the proposed circuit in the conclusion, specifically addressing the requirements of fine-tuning. The updated conclusion highlights the importance of the fine-tuning process for achieving optimal performance. In addition, research directions such as reduced tuning sensitivity and automated tuning methods have been outlined for the circuit's usability and scalability.

“Meanwhile, the proposed CTLA requires fine-tuning process during the circuit design, which is quite essential to guarantee optimal performance and provide flexibility across various application fields. Thus, it would be a future work to reduce tuning sensitivity and explore automated tuning methods, thereby enhancing the usability and the scalability of the circuit. Despite these challenges, the proposed CTLA has a potential to broaden its applications further and hence impact the fields of LiDAR technologies.”

4. In Table 1, there is an issue with the alignment of the first column.

(ans.) It is corrected.

Reviewer 2 Report

Comments and Suggestions for Authors

The paper describes a A Low-Noise CMOS Transimpedance-Limiting Amplifier for LiDAR systems.

The manuscript is well written and shows the advantage of using the proposed TIA with passive and active feedback. The simulated results demonstrate the idea and the performance comparison with the state-of-the-art ADCs is very good. The experimental results further enhance the advantage of the proposed method.

The manuscript however needs some additional information.

1 - How does the stability of the TIA behaves with mismatch. Is missing a Monte Carlo analysis of the solution.

2 – Please provide the corner simulations for the system which demonstrates the robustness of the design.

3 – How many samples were tested. This information is not provided.

Reviewer 3 Report

Comments and Suggestions for Authors

1) The short-range LiDAR sensor system must be included in a figure and some details about this application must be included. 

2) Please add a clear justification why the used area is fully reduced compared with other solutions in Table 1. 

3) The dynamic range (dB) of the proposed solution is 56.6 dB, while the previous work in [16] has a dynamic range (dB) of 50dB. How is ensured the  Dynamic Range Extension with the proposed solution if only 6dB is the improvement with respect to the literature.

4) Please comment about the bandwidth responses of diodes and APD in Fig. 2.

5) Why 180nm are used for this application? Please, add the main justification to use an old PDK with respect to for example 65nm.

6) The ring oscillator in Fig. 3 must be well designed. Why only three steps are used?

Author Response

1. The short-range LiDAR sensor system must be included in a figure and some details about this application must be included.

(ans.) Figure 1a shows the short-range LiDAR sensor system. We have included some details about this application in the revised manuscript (line 35 – 41).

2. Please add a clear justification why the used area is fully reduced compared with other solutions in Table 1. 

(ans.) Ref. [18] incorporated a TIA, a timing discriminator, bias circuits, and a threshold voltage generator. These led to a large area of 1.2 × 1.13 mm2 including I/O PADs. Ref. [19] included a 16-channle array that occupied the area of 5.0 × 1.1 mm2 in total. Ref. [20] had the core area of 0.55 × 0.3 mm², excluding I/O PADs. However, this is still larger than that of the proposed CTLA (= 0.18 × 0.05 mm²), because Ref. [20] included the power-on calibration circuits and the resistor arrays in the chip. Ref. [21] occupied the area of 1.1 × 0.42 mm², which included an on-chip APD followed by a multi-block architecture (especially with a differential TIA. We reckon that the proposed CTLA in this work results in a reduced core area because of its much simpler architecture, as shown in Fig. 1b.

3. The dynamic range (dB) of the proposed solution is 56.6 dB, while the previous work in [16] has a dynamic range (dB) of 50 dB. How is ensured the Dynamic Range Extension with the proposed solution if only 6 dB is the improvement with respect to the literature.

(ans.) In general, dynamic range is measured by the range of input currents that a TIA can handle with no significant distortions. Therefore, 6-dB improvement indicates that the range of the input currents can be doubled, which is quite significant. In particular, the minimum input current is solely determined by the noise performance of the TIA, and it is well known that the noise reduction is very challenging. As shown in the Table 1, the minimum detectable input current of this work is less than a half of Ref. [16]. This 6-dB improvement is noteworthy.

4. Please comment about the bandwidth responses of diodes and APD in Fig. 2.

(ans.) Typically, the bandwidth of optical devices (either photodiode or APD) should be designed to be much wider than that of the receiver bandwidth. In this work, we have utilized an on-chip APD (with 40-µm diameter of its optical window) that provides the -3dB bandwidth of 1.7 GHz under a reverse bias voltage of 10.25 V [17].

5. Why 180nm are used for this application? Please, add the main justification to use an old PDK with respect to for example 65nm.

(ans.) The realization of a low-cost CMOS solution is our main purpose of this work. Therefore, a 180-nm CMOS is preferred to 65-nm because of its low-cost. Moreover, the efficient photo-response of the on-chip APD mandates a smaller number of metal layers in CMOS processes, thus preferring a 180-nm CMOS (with less than 6 metal layers, 4 layers preferred for better response) to a 65-nm CMOS (with 8 metal layers).

6. The ring oscillator in Fig. 3 must be well designed. Why only three steps are used?

(ans.) Odd-number stages are required to acquire the negative feedback loop. Usually, a three-stage structure is preferred to a single-stage because of its larger voltage gain. Yet, a five-stage architecture is not desirable because an undesired delay might occur in the closed feedback loop.

Reviewer 4 Report

Comments and Suggestions for Authors

The manuscript titled: “A Low-Noise CMOS Transimpedance-Limiting Amplifier for Dynamic Range Extension” represents the original work of the authors.

The abstract is well-structured and provides a detailed overview of the research. A detailed overview and discussion of published research in this area is given, as well as what has been added to this area is presented. A goal of the research is shown and measuring and methodological description is presented. The manuscript is focused on a low-noise CMOS transimpedance-limiting amplifier designed for LiDAR sensor systems. The introduction of a dual-feedback architecture combining passive and active feedback for automatic limiting operations is a novel concept and is well-articulated. It is particularly significant that the findings are related to real-world applications, emphasizing the importance of energy efficiency in LiDAR systems.

Although some figures should be improved, the results are clearly presented in this manuscript.

The conclusions are consistent with the presented results, evidence and arguments. Also, they deal with the main question posed. In addition, the references used are appropriate and, moreover, most of them have been published in the past few years.

Considering that interesting research is presented the paper should be accepted (after minor corrections).

Some suggestions to the authors:

-        On page 6, in line 174 there is reference [12]. However, previous reference is [7] on the page 2. References [8] – [11] are not mentioned in the text. They must be mentioned in the text, given that they are in the list of references.

-        Fig. 15 and Fig. 17 should be somewhat larger.

-        With specific performance metrics (e.g., power savings percentage, or dynamic range improvement) would strengthen the impact and provide tangible evidence of this CTLA's advantages. Also, a solider emphasis on the potential broader impact of this manuscript, beyond short-range LiDAR sensors, might enhance its importance. Also, it would be useful to mention and wider importance of LiDAR technology. Due to the aging population, the social aspect of the application of these systems is increasingly significant.

-        After eq. (2), at the beginning of line 137 “,” should be removed.

-        The font size of the labels on the charts in Figure 5a should be increased and then these charts can be reduced.

-        Minor editing of English language required. Overall, the paper itself can be improved by correcting minuscule grammatical errors, such as missing articles.

-        Some typos should be corrected.

Author Response

1. On pp. 6, line 174: there is reference [12]. However, previous reference is [7] on the pp. 2. References [8] – [11] are not mentioned in the text. They must be mentioned in the text, given that they are in the list of references.

(ans.) We have corrected the manuscript to include the references.

2. Fig. 15 and Fig. 17 should be somewhat larger.

(ans.) We have reduced the sizes of Fig. 15 and Fig. 17.

3. With specific performance metrics (e.g., power savings percentage, or dynamic range improvement) would strengthen the impact and provide tangible evidence of this CTLA's advantages. Also, a solider emphasis on the potential broader impact of this manuscript, beyond short-range LiDAR sensors, might enhance its importance. Also, it would be useful to mention and wider importance of LiDAR technology. Due to the aging population, the social aspect of the application of these systems is increasingly significant.

(ans.) In Section 3, we have described the advantages of the proposed CTLA in the manuscript, such as low power consumption, wide dynamic range, less distorted pulse response, etc. These performance metrics were supported by both simulations and measured results.

4. After eq. (2), at the beginning of line 137 “,” should be removed.

(ans.) We have removed the comma.

5. The font size of the labels on the charts in Figure 5a should be increased and then these charts can be reduced.

(ans.) We have corrected the font size and the charts as commented.

6. Minor editing of English language required. Overall, the paper itself can be improved by correcting minuscule grammatical errors, such as missing articles.

(ans.) We have conducted thorough review of the manuscript to correct grammatical errors and missing articles.

7. Some typos should be corrected.

(ans.) We have corrected typos.

Round 2

Reviewer 3 Report

Comments and Suggestions for Authors

1) Figures can be improved in format and resolution. This point will be increase the impact of this research.

2) A reference about preference of 180nm, 130nm over 65nm, 28nm must be included.

Author Response

1. Figures can be improved in format and resolution. This point will increase the impact of this research.

(ans.) We have improved the figures.

2. A reference about preference of 180nm, 130nm over 65nm, 28nm must be included.

(ans.) We have added the following description and the corresponding reference in the manuscript.

“While smaller process nodes such as 65 nm or 28 nm offer advantages in terms of power efficiency and speed, their increased design complexity, process variability, and manufacturing costs often make larger nodes like 180 nm or 130 nm more suitable for certain applications, especially those requiring high stability and lower design complexity [18]. In particular, the efficient photo-response of the on-chip APD mandates a smaller number of metal layers in CMOS processes, thus preferring a 180-nm CMOS.”
